# A mapping review of worldwide current and previous cohort research programmes in cats and dogs

Jessica Irene Landolt[1], Dan G. O'Neill[2], Stefan Unterer[1], Sonja Hartnack[3], Malwina Ewa Kowalska[3,4]*

1 Clinic for Small Animal Internal Medicine, Vetsuisse Faculty, University of Zurich, Zurich, Switzerland, 2 Pathobiology and Population Sciences, The Royal Veterinary College, Hawkshead Lane, North Mymms, Hatfield, Herts, United Kingdom, 3 Section of Veterinary Epidemiology, Vetsuisse Faculty, University of Zurich, Zurich, Switzerland, 4 Section of Ophthalmology, Vetsuisse Faculty, University of Zurich, Zurich, Switzerland

* malwina.kowalska@uzh.ch

## Abstract

Cohort research programmes follow individuals over time to enable study of effects from various factors on health or other outcomes. To date, the global distribution of formal cohort programmes in cats and dogs has not been mapped, and a comprehensive synthesis of their methodological characteristics is lacking. That limits methods improvement and wider adoption of cohort programmes in veterinary medicine. A mapping review methodology aligned with the JBI Manual for Evidence Synthesis was used to summarise existing cohort research programmes on cats and dogs worldwide. Electronic databases were searched (Embase, MEDLINE, Scopus, Web of Science) to identify eligible cohort papers, followed by a two-step selection process (title and abstract screening, full text screening) for paper inclusion. Information extracted at the individual cohort programme level covered: 1. location and veterinary specialty; 2. study design; 3. study variables; 4. collected data; 5. recruitment and retention strategies. Database searches yielded 6,777 unique papers, of which 73 met the inclusion criteria. Twenty-two programmes were identified, predominantly in the UK (8/22, 36%) or US (6/22, 27%) with 55% (12/22) involving dogs. Most of the programmes (18/22, 82%) aimed at disease prevention. Out of 19 programmes for which full-text papers were available, and therefore more information could be extracted, animal demographics were the most commonly considered study variable (15/19, 79%). The biggest reporting gaps were identified in the study planning phase, design, and programme management. Consequently, limited information was retrievable from the programmes papers to create learning opportunities for other researchers planning future cohort programmes. Improved or new reporting practices are needed to enhance knowledge sharing and promote cohort programmes in veterinary medicine. The study protocol was preregistered on the 27th of December 2023 (https://osf.io/wkg53/).

**Data availability statement:** Data used for all analyses, analytic R code, and screener instructions are available at the OSF (https://osf.io/wkg53/) (DOI: 10.17605/OSF.IO/WKG53).

**Funding:** Dr. Stefan Unterer received financial support from the Stiftung für Kleintiermedizin for "The Growing Dog Project," Tierspital, University of Zurich.

**Competing interests:** The authors have declared that no competing interests exist.

## Introduction

Cohort research offer immense potential for health advancements in both human and veterinary medicine. These studies follow individuals over time to generate temporal evidence exploring associations between exposures and health (and other) outcomes [1]. Digitalisation has increased the opportunity for large-scale cohort research, often now considered more as research programmes than just a single study due to their complexity and multipurpose applications. These cohort research programmes generally collect a multitude of data, typically from individuals followed throughout their lives, that can enable answering a wide range of research questions [2]. Unlike a one-off cohort study, which is also longitudinal, but can be either prospective or retrospective, and assess risk of developing a specific condition (typically one research question) [3].

The growing public expectation for better veterinary care and interest in preventive veterinary medicine, along with increasing numbers of pets and the alluring potential use of pets in their home environment as models for human disease, all combine to create a high perceived value from cohort research programmes in veterinary medicine [4]. Domestic cats and dogs, whose lives are closely entwined with humans and often exposed to similar environmental factors, offer the research advantage of developing disease over shorter timespans than humans, making them desirable subjects for translational studies [5].

While the potential for important discovery from cohort research programmes in veterinary medicine is clear, the temporal, organisational, ethical and financial requirements are substantial and may deter some researchers [6,7]. However, experiential learning gained from existing cohort research programmes could offer many insights on study design and pitfalls that could mitigate some of these challenges. Unfortunately, the scarcity of veterinary-specific cohort methodological manuscripts often forces veterinary researchers to rely heavily on human literature that may be poorly generalisable to the veterinary sphere because of differences in design features such as owner consent forms and veterinary clinical research ethical considerations [8,9].

At present, the veterinary research community lacks a large-scale, systematic overview of existing veterinary cohort programmes. This lack hinders deeper understanding of the current state of veterinary cohort designs, impedes efforts to improve their conduct, reporting, and lasty delays wider adoption of cohort programmes. Pugh et al. [10] consolidated information on some existing cohort programmes (focusing on dogs only) and advocated for their expanded future application. A systematic consolidation of the available literature on cohort programmes in cats and dogs could guide researchers by providing necessary insights and veterinary-specific considerations to enhance the success of new future cohort research programmes.

A mapping review systematically identifies, categorizes, and synthesizes existing research to provide a comprehensive overview of a particular field [11]. Unlike systematic reviews, which focus on answering a specific research question often framed in the PICO (Population, Intervention, Comparison, Outcome) format, a mapping review addresses broader research questions. They do not assess the quality of the included studies in-depth but rather focus on the quantity, types, and characteristics of research

available [12]. For the current work, we chose to apply a mapping review methodology because it allowed for gaining a better understanding of the broader landscape of veterinary cohort programmes, enabling us to synthesize existing studies, identify gaps in research, and present the current state of veterinary cohort research on cats and dogs in an accessible format.

With the overarching aim of reporting the first systematic summary of cohort programmes specifically in cats and dogs to support the community of researchers planning or already conducting similar cohort programmes, we used a mapping review methodology and formulated following research questions:

### Primary research questions

RQ 1: In which countries have veterinary cat and/or dog cohort programmes been implemented, and when did they start?
RQ 2: Which veterinary research specialties were most frequently represented in the publications from cohort programmes?
RQ3: What are the methodological features of each cohort programme: (A) study design aspects (pre-registration, ethical clearance, owner consent, reported power calculation, and samples size), (B) patient enrolment criteria, (C) considered study variables and primary outcomes, (D) data collection methods, (E) source of collected data and (F) recruitment methods and retention strategies.
RQ4: If mentioned in the publications from each cohort programme, what were the drop-out rate and the follow-up period?

### Secondary research question

RQ5: Does the funding structure affect the planned target sample in cohort programmes?

## Material and methods

### Protocol and pre-registration

The protocol was drafted using JBI Manual for Evidence Synthesis [13] and was preregistered with the Open Science Framework (OSF) on the 27th of December 2023 (https://osf.io/wkg53/). Results are reported according to PRISMA-ScR (checklist available at OSF https://www.prisma-statement.org/scoping).

### Eligibility criteria

For the mapping review, we distinguished a cohort research programme from a traditional cohort study. For the mapping review, a cohort research programme was defined as any form of longitudinal data collection that follows a group of individuals over time with the potential to assess associations between a range of potential considered study variables and outcomes that were not required to be stated in advance when setting up the data collection. These research cohort programmes are designed to collect a wide range of data types and/or large quantities of data. A key distinction from one-off cohort studies is that cohort research programmes are designed to allow multiple research questions to be addressed over time rather than focusing on a single exposure-outcome relationship (traditional cohort study) [14]. In this mapping review, we specifically focused on cohort research programs involving only cats or dogs, which we will refer to as "programmes".

### Inclusion criteria

In the eligible programmes, animals are enrolled primarily based on demographics (e.g., cat or dog species, age, sex, breed) rather than health status (e.g., cancer registries), and data are collected at the individual animal level. For the current review, only research results from a programme with privately-owned cats or dogs or that described such a cohort programme design were eligible and only primary research or study protocols with complete texts. The year of the paper publication and language were not restricted.

## Exclusion criteria

Cohorts that focused exclusively on validating methods or establishing laboratory reference values, that collected data over time for only one health outcome, that did not collect either cat or dog data, or that only included animals kept in laboratory conditions were excluded.

## Information sources

The Medline with PubMed, Embase, Web of Science, and Scopus databases were searched from their inception up to the search date of 19th December 2023. These four chosen databases include most of the classical veterinary journals [15]. Grey literature (i.e., produced on all levels of government, academic, business and industry in print and electronic formats, but not controlled by commercial publishers [16]) was not sourced but an experienced veterinary epidemiologist (DON) was interviewed to identify further programmes beyond those included based on databases search.

## Identification of relevant papers

The online database search strings were developed in consultation with the current research team by an experienced librarian using a sample of three eligible papers (their keywords, titles. and mesh terms). We validated the search strategy by confirming that three other papers, previously deemed by us as eligible, were retrieved using the search string. The final search string for the mapping review is available in S1 File. Subsequently, the search results were imported into Rayyan [17], and de-duplicated.

## Selection of sources of evidence

Evidence selection involved two-stages (Stage I – title/abstract screening, Stage II – full-text screening). Until the completion of Stage II, the screened results are referred to as 'records' because they may not include the full text (e.g., posters or conference abstracts). After passing Stage II, they are classified as "papers".

**Stage I – titles/abstract screening.** To optimise the selection process, two of the authors (JL and MK) firstly independently reviewed titles/abstracts for 50 records. These reviews were masked to each other's decisions. Any disagreements between the two reviewers were discussed and resolved, and updated questions were formulated to promote a more consistent selection process going forward (below). Keywords used in Rayyan for inclusion can be found in S1 File. A semi-automated screening (AI-assisted method) was used that involved sorting titles/abstracts by ranking them after every 50 records were screened.

Rayyan's AI-powered Relevance Ranking learns patterns of decision criteria as reviewers make decisions on inclusion/exclusion and uses that learning to rate the probability that a record will be included. Records with highest ranking are moved to the top of the screening list.

In Stage I, records were considered eligible if questions 1–3 were answered "yes," and questions 4–6 were answered "no."

1. Does the record describe primary research or a study protocol involving cats or dogs that were not enrolled based on their health status alone?

2. Was at least part of the data for this study collected after the animal inclusion in the study?

3. Were data collected over a period of time, with potentially more than one time point for each animal, and with the epidemiological unit being at the level of individual animal?

4. Did the study involve solely applying an intervention (such as surgery, treatment, therapy, or education of pets guardian) to at least some of the animals in the study (study arm)?

5. Did the study solely collect data with the intention of answering only one research question?

6. Did the study solely aim to validate a set of methods or to establish some laboratory reference values?

**Stage II – full text screening.** Records without full texts were excluded. JL reviewed the full texts of all records, applied the six eligibility questions again (this time to the full text) and performed data extraction on all the papers that passed Stage II selection based on the six selection questions.

## Extraction and charting of data

Using a random sample of 4 papers selected from the papers that made it through Stage II selection, a data collection sheet drafted in advance was piloted and revised. Data extraction from each paper was carried out by JL using Google Sheets [18]. A detailed data collection sheet is available at S1 File. In brief, the extracted items included:

- Paper metadata: authors and their affiliations, institutions, year of publication, journal specifics, Digital Object Identifier (DOI), study title, and abstract (RQ1).

- Number of published papers per programme identified by the mapping review search and the actual number of papers published per programme found through additional searches such as programmes websites, papers references and social media (RQ1).

- Enrolment status (whether or not programme is still enrolling animals) defined based on the included papers or information on the programme's website (as of 5th of July 2024).

- The geographic scope of the programme – 1. the countries where data were collected, 2. the institutions and organisations involved based on the first and last authors affiliation (RQ1).

- The veterinary research specialty(s), classified according to the Textbook of Small Animal Internal Medicine"by Stephen J. Ettinger and Edward C. chapters listed in S3 File [19] (RQ2).

- The stated core aims of the research categorized into disease prevention, control, diagnosis, treatment, or other (RQ2).

- Study design aspects including ethics approval statement (present/absent), pre-registration or registered report (reported/not reported), owner consent (statement reported/not reported; additional information about owner consent was provided), power calculation (reported/not reported), sample size estimation (reported/not reported). We focused solely on whether the power and sample size calculations were reported, but not on their appropriateness, correctness or reproducibility (RQ3).

- Animal enrolment criteria (species, breed, sex, age, other) (RQ3).

- Considered study variables categorized into animal demographics, guardian of pets demographics, physical activity, environmental factors, animal behaviour, diet, medication, and health status (RQ3).

- Considered primary outcomes: how many and which (RQ3).

- Data collection methods: frequency and whether a data collection template was predefined. (RQ3)

- Source of collected data (the origin of the information used in the programme), categorized into questioner, electronic health records, biological samples, and others which were identified after reading the papers.

- Recruitment and promotion methods used. (RQ4)

- Retention strategies and incentives. We believe that covering the costs of diagnostic tests and veterinary visits partially or fully can potentially incentivize the guardians of pets to participate in the study [20,21] (RQ4).

- Planned programme target sample, total number of recruited individuals and study population that matched inclusion criteria during the study duration (RQ5).

- Funding sources; categorized as philanthropic private donations either direct or via non-governmental organization, industry, government, and university. (RQ5).

### Collation, summation, and reporting of results

The included papers were grouped by programme. Descriptive statistics were performed at the programme level using the statistical software R Studio (version 4.4.0) [22]. Continuous variables were summarised as median, interquartile range (IQR) and a range between minimum and maximum. JL searched PubMed [23] programme websites, and social media to extract the total number of papers per programme (beyond those identified by the mapping review search).

To address RQ5 about the funding structure and how it affects the programme planned target sample, an exploratory analysis was planned. It included calculating median, 95% confidence intervals and using a box plot to visually compare the distribution of planned target sample across different funding sources. The planned target sample was often not reported (see Table 3). Therefore, the total number of recruited individuals and study population that matched the inclusion criteria for each programme was additionally extracted from papers. The highest number reported in the papers was utilized for analysis at the programme level. To explore RQ5, we divided programmes into two categories: 1. at least one funding source classified as industry; 2. no industry involvement.

### Deviations from the original protocol

In the course of the mapping review, several adjustments were made to the original project protocol:

(1) Terminology update: shifted from referring to cohort "projects" to "programmes" to better encompass the complexity of these initiatives. These programmes are not merely small projects but consist of diverse layers of work and multiple components over time [24]. We believe that the term "programme" more accurately reflects the depth of their complexity.

(2) Change in grey literature search: Unfortunately, pre-registration of veterinary research cohort programmes or studies is still not widely used. The search of "cohort" and "longitudinal" in the preclinicaltrials.eu database yielded no results that met our inclusion criteria. Therefore, we decided to interview an expert to identify additional programmes.

(3) Inclusion criteria refinement: During the selection process, we encountered many cohort studies that were primarily focused on method validation, establishment of reference range values, or data gathering for only one outcome. These studies were outside the interest of the mapping review, and therefore, we refined our inclusion criteria to exclude those reports.

(4) To answer RQ5, in programmes where the planned target sample was not reported, we used the highest number of animals reported in one of the papers (either the total number of recruited animals or study population), assuming that the reported number of animals was what research budget allowed for. This approach was necessary because many papers lacked information on the planned target sample.

## Results

### Characteristics of included papers

The literature search yielded 6,777 records after deduplication for entry into Stage I. Of the 137 that entered Stage II selection, 3 records were excluded as duplicates, and 61 records were excluded because the full text was not available or they failed at least one of the six selection questions. This left a final 73 papers eligible for inclusion in the mapping review

(Fig 1). These 73 papers were mapped to 21 individual cohort research programmes (Table 1). One additional programme was identified and subsequently included in the mapping review based on the interview with an expert (DON).

Based on the information from the included papers and programme websites, the start of animals enrolment ranged from 1994 to 2022; 12/22 (54%) programmes were still enrolling animal patients [ID: 1–8, 12, 20–22], 4/19 (18%) were considered as terminated [ID:10, 11, 14, 18], and it was not possible to establish the current status for the remaining 6/22 (27%) [ID: 9, 13, 15–17, 19]. The median number of papers per programme identified by the mapping review search was 2 (IQR: 1–4, range 0–14); when including those identified by the additional search (described in Extraction and charting of data: Number of published papers), median was 15 (IQR: 3.57–35.5, range 1–141).

Two programmes [ID:16, 18] that were only identifiable from conference abstracts and one [ID:22] for which papers were not retrieved with a search string but through an expert interview were excluded from detailed analysis beyond RQ3.

### The geographic scope of cats and dogs cohort research programmes (RQ 1)

Overall, 14/22 (61%) of the cohort programmes were run from institutions and organisations (based on the first and last author affiliation) based in the UK (8/22, 35%) or the US (6/22, 26%) [ID UK: 1, 5–7, 12, 13, 21, 22 and ID US: 2, 3, 8, 15, 17] (Table 1). For 20/22 (91%) programmes, the geographical location of the responsible institutions and organisations

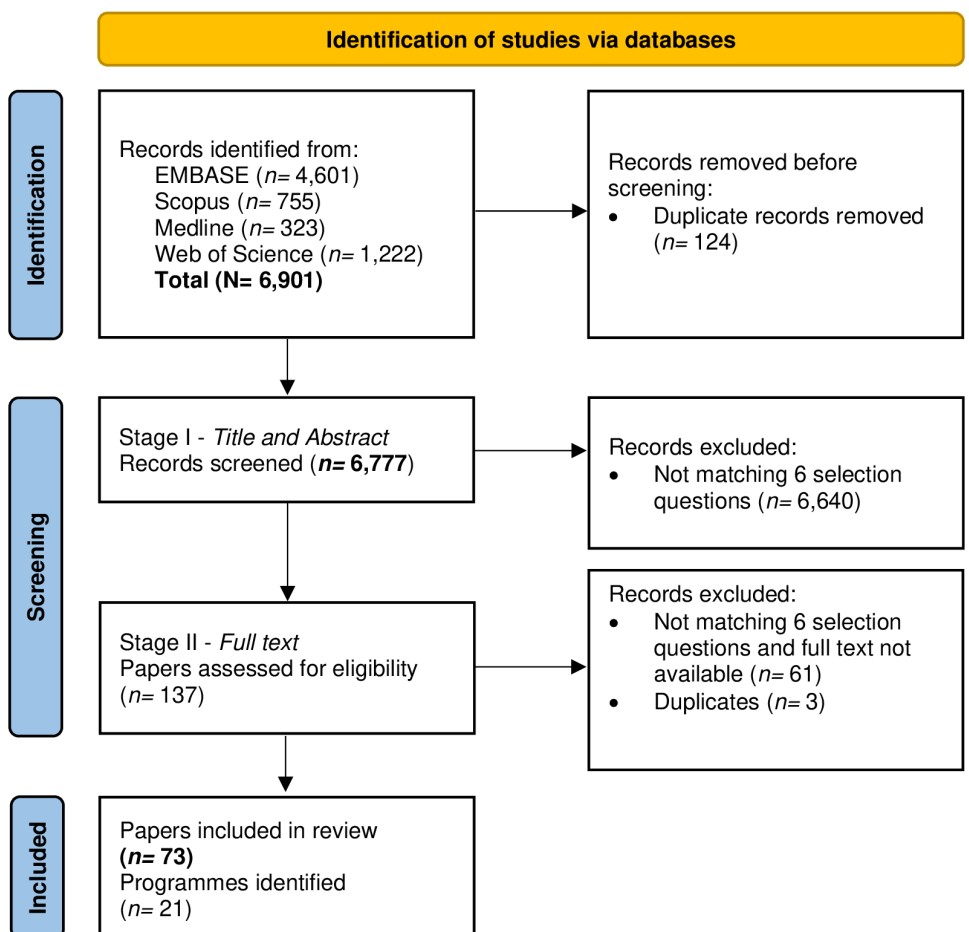

**Fig 1. PRISMA flow diagram of records selection for the cats and dogs research cohort programmes mapping review.**

**Table 1. A summary of cats and dogs cohort research programmes selected for the mapping review, along with papers retrieved via a literature search.**

| IDª | Cohort Research Programme name | Researched Species | URL | Papers belonging to the same programme |
|---|---|---|---|---|
| 1 | Generation Pup | Dog | https://generationpup.ac.uk/ | [25–30] |
| 2 | Dog Aging Project | Dog | https://dogagingproject.org/ | [31–41] |
| 3 | Golden Retriever Lifetime Study – Morris US | Dog | https://www.morrisanimalfoundation.org/golden-retriever-lifetime-study | [42–45] |
| 4 | VetCompass Australia | Cat & Dog | https://www.vetcompass.com.au/ | [46] |
| 5 | VetCompass UK | Cat & Dog | https://www.rvc.ac.uk/vetcompass/about/overview | [47–60] |
| 6 | Bristol Cats | Cat | https://www.bristol.ac.uk/vet-school/research/projects/cats/#:~:text='Bristol%20Cats'%20is%20the%20cat,region%20in%20the%20early%201990's. | [61–67] |
| 7 | Dogs life | Dog | https://www.dogslife.ac.uk/ | [68–71] |
| 8 | Mars Petcare | Cat & Dog | https://marspetcarebiobank.com/ | [72] |
| 9 | CaniAge | Dog | None found | [73] |
| 10 | TeamMate | Dog | https://vetlife.co.nz/farm-vet/working-dog-vet/teammate-working-dog-project/ | [74–76] |
| 11 | West Highland White Terrier | Dog | None found | [77,78] |
| 12 | CatPAWS | Cat | https://www.facebook.com/ageingcats | [79] |
| 13 | C.L.A.W.S. | Cat | https://www.bristol.ac.uk/vet-school/research/projects/claws/https://www.sciencedirect.com/science/article/pii/S0167587717301320?via%3Dihub | [80] |
| 14 | Norwegian Large dogs | Dog | None found | [81–85] |
| 15 | French Frailty | Dog | None found | [86,87] |
| 16 | Canadian K9 (LYME) study | Dog | Page no longer available | [88] |
| 17 | Mid Missouri dogs | Dog | None found | [89] |
| 18 | Cohort study western Massachusetts | Dog | None found | [90] |
| 19 | het Boxerproject | Dog | https://dspace.library.uu.nl/handle/1874/1149 | [91,92] |
| 20 | Agria Pet Insurance (Research programmes) | Cat & Dog | None found | [93–95] |
| 21 | Pandemic Puppies | Dog | https://www.rvc.ac.uk/vetcompass/research-projects-and-opportunities/projects/rvc-pandemic-puppies-survey | [50,96] |
| 22 | Small Animal Veterinary Surveillance Network (SAVSNET) | Cat & Dog | https://www.liverpool.ac.uk/savsnet/ | Identified via expert interview |

Legend:

ªReference ID of the programme identified by the mapping review literature search on expert interview.

corresponded to the location of data collection. It was unclear where the animals were enrolled and where the data was collected in the remaining two programmes. Figure 2 represents the geographic scope of the 20 programmes that could be assigned to a specific country.

The median time from the start of enrolment to the publication of the first paper was 5 years (IQR 3–9, range 1–26). Only two programmes included a paper that clearly stated the date when the planning phase started [ID: 1, 2]. For ID 1, the duration from the start of the planning to the first publication was 3 years and for ID 2, this duration was 8 years. The duration from the start of the planning to the first patient enrolment was 1 years for ID 1 and was 4 years for ID 2. S1 Fig presents year of enrolment start and first publication (Fig 2).

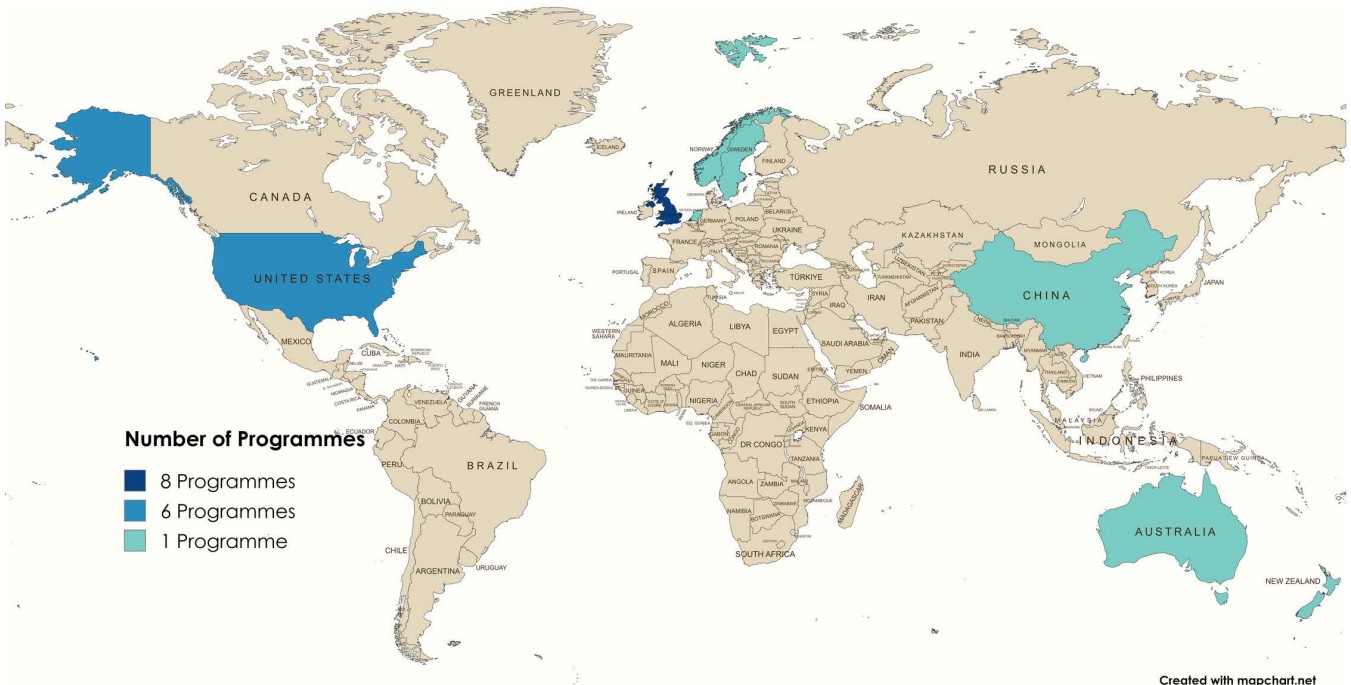

**Fig 2. Map illustrating the global distribution of the cats and dogs cohort research programmes selected for the mapping review, with the shades indicating the number of programmes per country.** Reprinted from www.mapchart.net under a CC BY 4.0 license, with permission from the owner and developer of MapChart.

### Veterinary specialties and programmes core aim (RQ 2)

Veterinary research specialties represented in cohort programmes are presented in Fig 3. Overall, 11/22 (50%) programmes were classified as preventive medicine speciality [ID: 1, 4–8, 10, 14, 18–20], 4/22 (18%) as geriatrics speciality [ID: 2, 9, 12, 15], and equal number of 2/22 (9%) as infectiology [ID: 16, 17] and behavioral medicine [ID: 13, 21]. Most of the programmes 18/22 (82%) aimed at disease prevention [ID: 1–8, 10–16, 19–20], with 1/22 (4%) serving as a pilot study for a human cohort [ID: 9], and 1/22 (4%) was described as a database for the surveillance and research [ID: 22]. The core aim of the research was unclear in the remaining 2/22 (9%).

### Study design aspects (RQ 3)

The results from here are presented for the 19 programmes with fuller information available and do not include the two programmes [ID: 16, 18] identifiable only from conference abstracts and one identified by an expert interview [ID: 22].

**Pre-registration.** Pre-registered reports were available in 2/19 (11%) programmes [ID: 2, 8]. For the remaining 17/19 (89%), neither information about pre-registration nor a registered report could be found. However, 8/19 (42%) programmes provided an overview of the cohort programme methodology and its organization together with preliminary results at a later stage [ID: 1, 3, 6, 7,10,11 13,19].

**Ethical approval.** In 12/19 (63%) programmes, a statement confirming ethical approval was provided in at least one paper identified via literature search.

**Owner consent.** A statement about the owner consent obtained was available for 13/19 (68%) programmes [ID: 1–8, 10–12, 14, 21]. Based on the information provided in the papers, the informed consent included offering opt-out (*n*=2) [ID: 4, 5], permission for the veterinarian to visit the premises (*n*=1) [ID: 10] and permission to be contacted by post/email (*n*=1) [ID: 7].

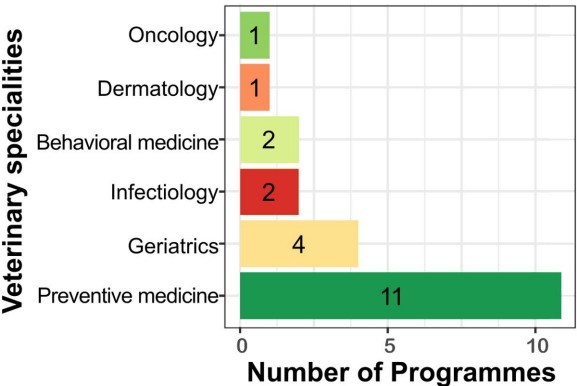

**Fig 3. Bar plot with the bar width representing the number of cats and dogs cohort research programmes, selected for the mapping review, in each veterinary research specialties.**

**Power calculation.** A power calculation description was available for 6/19 (31%) programmes [ID: 1, 3, 6, 8, 12, 14], with additional 1/19 (5%) programme reporting a separate power calculation for each paper [ID: 5]. For 12/19 (63%) programmes, a power calculation was not reported.

***A priori* sample size estimation.** In 6/19 (32%) programmes, an *a priori* sample size was specified [ID: 1, 2, 3, 8, 9, 12]. For 13/19 (68%) programmes, an *a priori* sample size was not reported.

### Enrolment criteria (RQ 3)

Dogs alone were included in 12/19 (63%) programmes [ID: 1–3, 7, 9–11, 14, 15, 17, 19, 21] and cats alone were covered in 3/19 (16%) programmes [ID cats: 6, 12, 13]. There were 4/19 (21%) programmes that covered both cats and dogs [ID cats and dogs: 4, 5, 8, 20]. Age was the most common enrolment criterion, with 10/19 (53%) programmes requiring animals to meet specified age criteria as part of inclusion criteria [ID: 1, 3, 6, 8, 10–14, 21]. Age was an enrolment criterion in all three programmes involving cats only [ID: 6, 12, 13] but was a criterion in 8/12 (64%) of the programmes focusing only on dogs [ID: 1, 3, 6, 8, 10–14, 21]. Detailed age inclusion/exclusion criteria across 19 programmes are available in S1 Table. Specific breeds were part of inclusion criteria in 5/19 (26%) programmes [ID: 1, 3, 7, 11, 14]. Breed as an inclusion criterion was often justified with the argument that a breed-specific population would be more homogeneous and therefor beneficial by reducing genetic variability and allowing for more precise and reliable study results. Individual breeds with a breed-specific programme were: Boxer [ID 19], Golden Retriever [ID 3], Labrador Retriever [ID 1], and West Highland White Terrier [ID 11]. One programme enrolled dogs from any of four breeds: Newfoundland, Labrador Retriever, Leonberger and Irish Wolfhound [ID: 14]. None of the 3 programmes that included only cats focused on a specific breed. Sex was not a factor considered for enrolment in any of the 19 programmes. Based on the text of included papers, we could not establish whether age, breed or sex were part of the inclusion criteria in the 3, 1, and 1 programmes respectively.

Additional enrolment criteria required by some programmes for dogs included being registered in a specified Kennel Club (*n*=2) [ID: 14, 7], available three-generation pedigree (*n*=1) [ID: 3], being a working farm dog (*n*=1) [ID: 10], and a puppy aged under 16 weeks purchased at any date during 2021 (*n*=1) [ID: 21]. Additional enrolment criteria required by some programmes for cats included living in the UK (*n*=1) [ID: 6] and guardian of pet willing to visit the research facility every 6 months (*n*=1) [ID: 12]. Additional enrolment criteria required in both cats and dogs programme included bodyweight ≥ 2.5 kg (*n*=1) [ID: 8].

### Considered study variables and primary outcomes (RQ3)

It was challenging to determine the complete list of study variables from the published papers considered for each programme. Some study variables may have been omitted from the papers, despite being collected in the programme. In

some cohort programme's papers, a review of the results sections revealed that certain study variables were reported despite not being listed in the methods section, complicating data extraction. Based on the information available in the included papers from the 19 cohort programmes, the following study variables were used: 15/19 (79%) animal demographics [ID: 1–8, 10–12, 14, 19–21], 12/19 (63%) used medications and preventives [ID: 1–8, 10–12, 14], 11/19 (58%) diet [ID: 1–3, 7, 8, 10–14], 11/19 (58%) environment [ID: 1–3, 6–8, 10–12, 19, 20], 11/19 (58%) guardian of pet demographics [ID: 1–3, 6–8, 10, 12–14, 21], 10/19 (53%) level of physical activity [ID: 1–3, 6–10, 12, 14], 9/19 (47%) and animal behaviour [ID: 1–3, 6, 8, 12, 13, 19, 21]. The median number of considered study variable groups per programme was 7 (IQR 5–8, range 3–8). Further details are presented in Fig 4.

Outcomes were analysed on the paper level (from 73 included papers). The median reported outcome per paper was 1 (IQR 1–1, range 0–11) (S2 Table). If papers reported more than one outcome, they could be grouped under a theme that connected them. Eight papers did not report any outcome and instead described developing instruments for data capturing, methods for data integration, or provided a detailed description of a programme.

## Data collection methods (RQ 3)

In 16/19 (84%) programmes, the data collection vehicle (e.g., clinical records query, questionnaire) was designed specifically for the study and the details were predefined [ID: 1–8, 10–14, 19–21]. Additional information about data management (e.g., format, storage, access) was available for the two programmes (11%) that published a registered report [ID: 2, 8].

Data collection frequencies varied across the programmes, with a notable trend of more frequent data collection during the earlier stages of the animals' lives. During the period of animals' lives aged 0–7 months, data collection took place six times (n = 1) [ID 7], five times (n = 1) [ID 19], four times (n = 1) [ID 1], three times (n = 1) [ID 14], and twice (n = 3) [ID 6, 8,

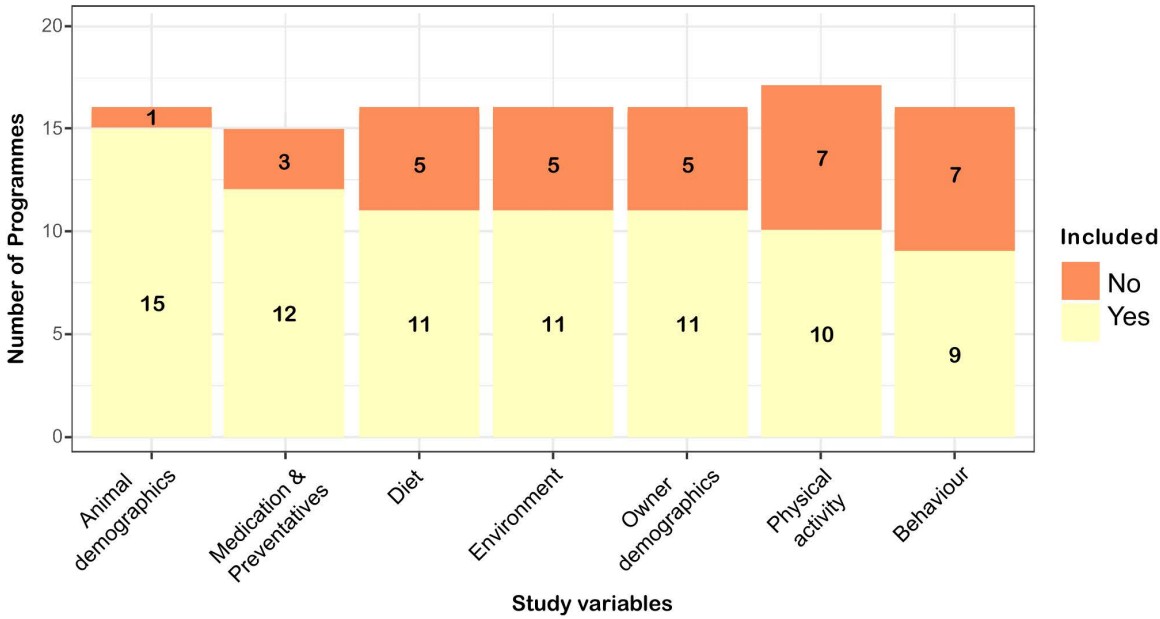

**Fig 4. Groups of study variables, considered in the selected cats and dogs cohort research programmes.** Bar plot illustrating the collected information on considered groups of study variables, indicated in yellow the number of programmes for which there was evidence that they were included and in orange the number of programmes for which they were not included. We could not determine whether study variables were used in some programmes. Therefore, the height of the bars does not correspond to 19, but to the total number of programmes for which information about study variables could be extracted.

13]. During the period when animals were aged 7–14 months, data collection frequency decreased, with programmes collecting data six times ($n=1$) [ID 7], three times ($n=1$) [ID 1], twice ($n=1$) [ID 8], but the majority collecting data once ($n=5$) [ID 6, 11, 13, 14, 19]. For animals older than 14 months, data collection was less frequent. In one programme, data were collected every three months [ID 7], in five programmes every six months [ID 1, 11, 12, 15, 19], but most programmes ($n=8$) collected data yearly [ID 2, 3, 6, 8, 9, 10, 13, 14]. Two programs collected data after each veterinary visit [ID:4, 5], regardless of animal age. For eight programmes (42%), the data collection frequency and intervals did not change over time [ID: 2, 3, 9, 10, 12, 15]. For three programmes (26%), based on the included papers, we could not establish the frequency and intervals of data collection.

## Sources of collected data (RQ 3)

There were three major sources of data: electronic health records 13/19 (68%) [ID: 1–8, 11, 12, 14, 19, 20], survey of pet guardians 13/19 (68%) [ID: 1–3, 6–8, 10–14, 19, 21], and biological samples 10/19 (53%) [ID: 1–3, 6, 8, 9, 11–14]. Eleven (58%) programmes used more than one source of data. For the 13 survey-based programmes, a digital format of the questionnaire was the most common (10/13, 77%). For the two web-based questionnaires, the paper format was also available upon pet guardian's request. For the biological sample programmes, samples were collected at one designated facility ($n=3$) [ID: 2, 8, 12], any primary care veterinary practice ($n=2$) [ID:3, 14], or at the pet guardians' home ($n=2$) [ID:10, 11], although the location or method of delivery could not be determined from the papers for two programmes. Biological samples were mandatory in 7/19 of programmes (37%) [ID: 3, 8, 9, 11–13, 14], and optional for an additional three programmes (16%) [ID: 1, 2, 6]. Faecal samples were the most frequently collected sample type ($n=9$) (Table 2). The data sources could not be determined for two programmes.

Other data sources used for programmes included: radiographs ($n=1$) [ID: 14], systolic blood pressure ($n=1$) [ID: 12], cat gingivitis score ($n=1$) [ID: 12], veterinary health cards (oral health, body condition score, assessment of heart) ($n=1$) [ID: 1], indoor dust samples ($n=1$) [ID: 11], pedigree ($n=1$) [ID: 19], necropsy results ($n=1$) [ID: 3], body condition scores and oral health scores cards filled out by the veterinarians ($n=1$) [ID: 6].

## Recruitment methods and retention strategies (RQ 3)

**Recruitment and promotion methods.** Convenience sampling was used in 17/19 of programmes (89%). In two programmes, animals included in the data analysis for publications were selected randomly from the pool of all animals who were patients in the veterinary practices collaborating with the cohort programmes [ID 4, 5].

**Table 2. Collected samples in ten cats and dogs cohort research programmes where biological samples were type of a data source.**

| Type of biological sample | Programme ID | | | | | | | | | | Total |
|---|---|---|---|---|---|---|---|---|---|---|---|
| | 1 | 2 | 3 | 6 | 8 | 9 | 11 | 12 | 13 | 14 | |
| *Faeces* | • | • | • | • | • | • | • | | • | | 8 |
| *Blood* | | • | • | | • | • | • | • | | • | 7 |
| *Hair* | • | • | • | • | | • | | | | | 5 |
| *Buccal Swab* | • | • | | • | • | • | | | | | 5 |
| *Urine* | • | • | • | | | • | | • | | | 5 |
| *Skin swab* | • | | | | | • | | | | | 2 |
| *Nail clippings* | | | • | | | | | | | | 1 |
| *Tears* | | | | | | • | | | | | 1 |
| **Total** | 4 | 5 | 5 | 3 | 3 | 7 | 2 | 2 | 1 | 1 | 34 |

The most frequently used method of promotion was through the programmes' own websites, accounting for 14/19 of programmes (73%) [ID: 1–8, 10, 12, 13, 19–21], with social media used in 8/19 (42%) programmes [ID:1–3, 6, 7, 12, 13, 21]. Most programmes (11/19, 58%) used multiple advertising strategies (Fig 5). There was an apparent trend that the more data sources were used, the more promotion methods were used (S2 Fig) For 5/19 (26%) programmes, information about promotion methods used to attract animal guardians was not available (Fig 5).

**Retention strategies and incentives.** Overall, 15/19 (79%) programmes did not disclose who covered the costs of required veterinary visits, analysis, or shipping of biological samples. Of the four (21%) programmes that did provide details, the pet guardian fully covered the costs in 3 programmes [ID:4, 5, 20] with a partial reimbursement offered in 1 [ID:3]. In two other programmes, guardians received gift vouchers unrelated to the veterinary visit that could be used for pet supplies [ID:10] or non-pet vouchers [ID:13]. Other strategies to promote owner engagement included sending monthly digital newsletters ($n = 2$) [ID:2, 7], awarding dogs with certificates of contribution to science [ID:1], and organizing digital competitions for pet guardians [ID:1].

### The drop-out rate and the follow-up period (RQ4)

Information on the planned duration was not reported in 6/19 (32%) of the programmes. A life-long cohort was planned in 7/19 (37%) of programmes. Four included dogs alone [ID: 1–3, 7], one cats [ID: 6], and two both dogs/cats [ID: 5, 6]. Participation for 10 years was planned in 1/19 (5%) [ID: 14], for 4 years in 1/1 (5%) [ID: 10], and for 3 years in 2/19 (10%) [ID: 9, 11]. Two of the planned lifelong cohorts stressed that the study duration depended on ongoing funding availability [ID: 1, 2] (Table 3).

### Funding type and programmes sample size RQ (5)

Information about the source(s) of funding was unavailable for 7/19 (37%) programmes. From the remaining 11 programmes, disclosed funding sources included that 8/19 (42%) used philanthropic private donations either direct or via non-governmental organization [ID: 1–3, 5–7, 13, 20], 6/19 industry (32%) [ID: 5, 6, 8, 10, 12, 20], 4/19 government (21%) [ID: 2, 3, 5, 21] and 1/19 university (5%) [ID: 10]. Six programmes (32%) reported more than one funding source. Due to the insufficient amount of data, we were unable to definitively determine the influence of industry funding and the amount of funding sources on the planned programme target sample (Fig 6). The median target sample for programmes supported at least partially by industry funding

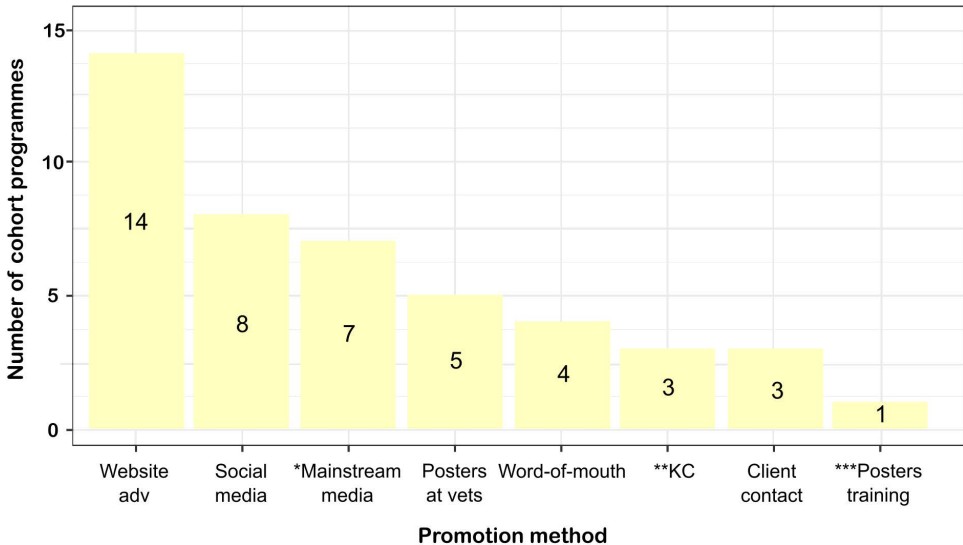

**Fig 5. Promotion methods used to encourage recruitment in the 19 cats and dogs research cohort programmes selected for the mapping review.**
*Mainstream media includes newspapers, television, radio, and magazines. **KC- Kennel Club. ***Posters training- posters displayed at dog training locations.

**Table 3. Number of enrolled animals and the status of cats and dogs research cohort programmes.**

| ID | Cohort research programme name | Specie | Enrolment | Planned target sample | Total recruited (year) | Study population[b] | Reported dropout |
|---|---|---|---|---|---|---|---|
| 1 | Generation Pup | Dog | Ongoing | Initially 5 000 updated to 10 000 | 3 686 (2020) 9 000 (2024)[a] | 718 (2020) 3 726 | 53% |
| 2 | Dog Aging Project | Dog | Ongoing | 100 000 | 50 000[1] | 27 541 | ni |
| 3 | Golden Retriever Lifetime Study – Morris US | Dog | Ended | 3 000/500 with cancer | 3 044 | 3 044 | ni |
| 4 | VetCompass Australia | Cat & Dog | Ongoing | opened | – | 2000 | ni |
| 5 | VetCompass UK | Cat & Dog | Ongoing | opened | – | 905 543 | ni |
| 6 | Bristol Cats | Cat | ended | ni | 2 203 (2017) | 1 816 (2017) | 17% |
| 7 | Dogs life | Dog | Ongoing | ni | 8 981 | 4 307 | ni |
| 8 | Mars Petcare | Cat & Dog | Ongoing[a] | 10 000cats 10 000 dogs | ni | ni | ni |
| 9 | CaniAge | Dog | Ended | 80 | ni | ni | ni |
| 10 | TeamMate | Dog | Ended | ni | ni | 641 | ni |
| 11 | West Hight White Terrier | Dog | Ended | ni | ni | 107 | ni |
| 12 | CatPAWS | Cat | Ended | 385 | 72 | 71 | 1% |
| 13 | C.L.A.W.S. | Cat | Ended[a] | ni | 396 | 300 | ni |
| 14 | Norwegian Large dogs | Dog | Ended | ni | ni | 647 | ni |
| 15 | French Frailty | Dog | Ended | ni | ni | 80 | ni |
| 17 | Mid Missouri dogs | Dog | Ended | ni | ni | 118 | ni |
| 19 | het Boxerproject | Dog | Ended | ni | ni | 2 629 | ni |
| 20 | Agria Pet Insurance (Research programmes) | Cat & Dog | Ongoing | ni | "Just over 600 000" | (649 matched exclusion criteria) | ni |
| 21 | Pandemic Puppies | Dog | Ended | ni | ni | 8040 | ni |

ni = no information available.

[a]Information retrieved from the programme's website or social media.

[b]Highest number among papers under the same programme. ni- no information available.

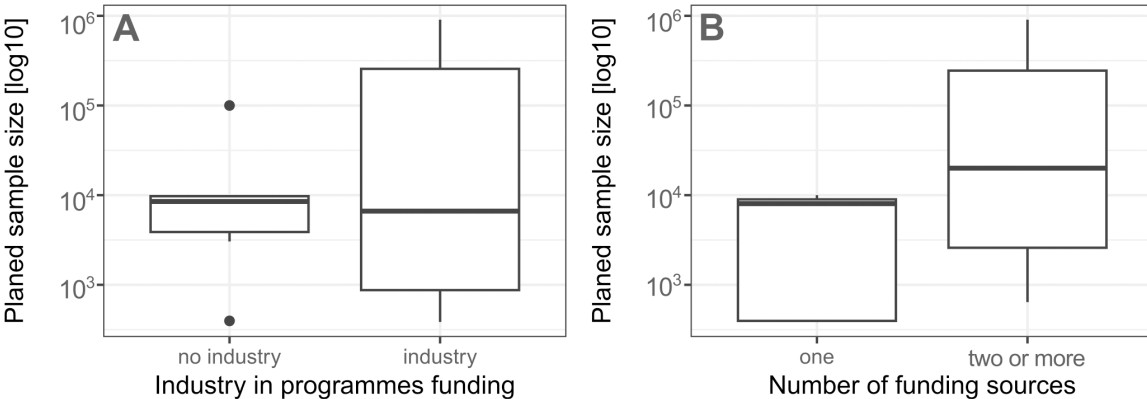

**Fig 6. Box plot representing planned target sample or the largest reported number of recruited animals and funding structure in the cats and dogs research cohort programmes selected for the mapping review.**

was 11 101 (95% CI: 385–90,545, $n=6$), while for those without industry support was 8,510 (95% CI: 396–100,000, $n=6$). The median target sample for programs supported by one funding source was 8,040 (95% CI: $-\infty$ to $\infty$, $n=5$), while for those with at least two funding sources was 20,000 (95% CI: 641–905,543, $n=7$). Note that program ID 5 had the largest number of animals participating among all programs and was co-funded by all four funding sources, including industry.

## Discussion

This mapping review identified and summarized 22 cohort programmes in an effort to assess the current state of cohort research in cats and dogs. Veterinary cohort programmes in cats and dogs remain relatively rare compared to the much higher frequency of cohort research in human medicine, as evidenced by the 120 oral health-related birth cohorts identified in just this one medical discipline in a similar scoping review of human literature [97]. The current mapping review set out five research questions, each of which will now be discussed separately.

### RQ 1: The geographical scope of cat and dog cohort programmes

There was a notable concentration of cohort programmes in English-speaking regions. The observed country distribution suggests that the high resource costs of cohort programmes heavily limits these to wealthier countries [98], where a strong pet-parent mentality may drive demand and resources for the advancement of veterinary services [4,99] while also supporting financial mechanisms like grants and established charities to support those programmes.

Interestingly, the geographical location of cohort programmes also reflects broader pet ownership patterns, with cat-only cohorts found exclusively in Europe, with none reported in China or the US. In Europe, cats are the most commonly owned household pets [100] while dogs outnumber cats in the US and China, with 44% of US households owning a dog compared with only 26% owning a cat [101].

Overall, 16/19 (84%) of cohort programmes included dogs, reflecting the veterinary industry's financial priorities, where canine medicine has greater economic value than feline medicine [102]. The higher existence of dog cohorts may also result from the proportion of dogs that are purebreed dogs with concerns about breed-related diseases, compared to cats, where domestic shorthairs dominate the population [103]. Additionally, the bond between humans and dogs is distinct from that between humans and cats, influencing various aspects of veterinary care and research priorities [104]. We conjecture that the unique bond between humans and dogs, along with a more organized dog community – comprising kennel clubs, charities, and similar organizations with a heavy focus on dog welfare – helps translate research questions into action and secure funding for cohort programmes. To increase the demand and effectiveness of veterinary cohort programmes for cats and dogs, efforts should focus on enhancing visibility and collaboration between regions to share successful strategies and lessons learned. Promoting these programmes in non-English-speaking regions and addressing financial barriers could help expand their reach and impact globally.

### RQ 2: Veterinary research specialties represented in the publications from cohort programmes

The majority of the programmes focused on disease prevention (18/22, 82%). The general focus of the cohorts aligned with the recent movement in small animal veterinary medicine from treatment to prevention [105]. However, without pre-registration, it is unclear whether individual programmes instituted planned shifts in focus over time or whether any changes emerged without prior planning later as new applications for the data were identified. Since our analysis relied on publications that may not reflect the original programme objectives, which could have evolved, e.g., to secure additional funding, pre-registration is recommended to ensure clarity and consistency.

### RQ 3: Methodological feature of cat and dog cohort programmes

Animal demography was the most frequently considered study variable while animal behaviour was the least common of the list in the current study. This distribution reflects a focus on study variables with well-established and easier

quantification methods. Although tools like the C-BARQ [106] are available, behavioural assessment is more complex during data collected than, e.g., simply asking whether an animal is male or female. Despite being challenging to quantify, including behaviour as a study variable, alongside data extracted from medical records, may allow for the discovery of novel disease development pathways [107,108]. This claim is supported by endocrinological studies in cats and dogs, where alterations in basal levels of thyroid hormones, glucose, or cortisol affect behaviour [109].

**RQ 4: Promotion method and retention of participants**

There is little published literature about what motivates pet guardians to commit to contributing to animal-related research. Surprisingly, social media was not the most frequently used promotion method to attract new participating guardians, with most programmes (14/19, 73%) relying on their own website for promotion. Programmes with diverse data sources appeared to use more advertising channels. Balancing the need for comprehensive data collection with the risk of overburdening and, therefore, discouraging participants seems challenging. While some programmes required minimal guardian involvement, e.g., those that collected electronic health records data directly from practice management system (PMS) such as VetCompass, others demanded high guardian participation, such as Golden Retriever Lifetime Study – Morris US that required collection of multiple biological samples at specific locations and over the lifetime of the dog. We noticed that data collected from young animals was gathered more frequently. However, the current mapping review is unable to conclude on which promotion and retention strategies are better.

**What limits learning based on existing cohort research programmes?**

Several gaps in the reporting of the methodology of cohort programmes were identified, which not only preclude the reproducibility of their results but also hinder learning opportunities for other researchers planning to set up a new cohort. Those gaps include:

(1) A paucity of information about the study planning phase. Only two cohorts provided information about this period.

(2) Not reporting dropout rates precludes an opportunity to compare and contrast the effectiveness of various retention strategies (when described).

(3) Absence of comprehensive information on owner consent and data management.

(4) Funding and other financial transparency was also generally lacking, with few programmes disclosing who covered the costs of veterinary visits and sampling. We would like to emphasize two programmes: one (ID:18) yielded (despite extensive additional searches) only an abstract informing the public about an intention to conduct a large cohort study, leaving uncertainty about whether the study was even conducted. The other (ID:8) originated from a large company, and only the registered report is accessible, with findings not available to the public.

**Possibilities to improve learning based on existing cohort programmes**

The findings from the current study suggest strong value for the veterinary research community to preregister studies (or to publish registered report), which could prevent selective reporting, foster reproducible science [110] and reduce the reporting shortcomings discussed in the paragraph above. Only 2 of the 22 programmes [ID: 2, 8] had published registered reports and it was notable that these were the programmes where the most detailed information about study design (e.g. information about the team) was available. Unfortunately, even well-reported programmes were still noted to miss out on sharing relevant information about establishing, implementing, and managing cohort programmes in dogs and cats that could help others who plan to establish their own cohort programmes. In the opinion of the authors, in addition to the items required STROBE-vet [111], publications should also share information about 1. the planning phase, 2. attachment of owner consent, 3. used retention strategies, 4. the funding sources, and 5. the costs that participants need to bear.

However, it is also relevant that word count limits for traditional scientific publications rarely allow enough opportunity to share the full set of information laid out here that would be useful for future programme design. Consequently, we recommend regular publication of fuller information on the programme level (e.g., on a programme website) that could include the total number of enrolled animals, currently participating animals, dropout rates, and programme findings.

### A cohort programme is a time investment

The current mapping review highlights that the planning and implementation phases of cohort programmes generally involve large time investment (see RQ1; 3 and 8 years). Pre-registration may be part of the planning phase. It is worth emphasizing that once the programme is initiated, good planning can "pay off" handsomely with a large number of publications of high quality and impact. For example, the VetCompass UK website reports having led to 141 publications from 2012 to 2023 (and S2 File), which have contributed to the improvement of clinical care and created training opportunities for undergraduate and postgraduate students [112].

### Digitalisation as an opportunity for cohort programmes

Observed improvement over time in data management was noted by the researchers during the current mapping review and this was ascribed to benefits from increased digitalisation. The papers published from programmes initiated between 1994 and 1998 reported major challenges related to manual data entry [ID 14, 19], while newer cohort programmes since 2010 have moved mainly to use electronic data collection platforms [ID 1, 2, 3, 7, 8] that made handling large amounts of data easier and cheaper. Further advances in digitalisation may enable real-time data entry via digital tools and mobile apps, which can minimize errors [71]. Digital data progress also supports global collaboration, since online data could be entered and accessed from anywhere in the world. Artificial Intelligence (AI) and machine learning should also enhance data extraction and inference from free-text [113].

### The mapping review limitations

We would like to acknowledge several limitations. First is the literature search string used in the mapping review. For example, none of the papers from the SAVSNET programme were retrieved through this process. To mitigate this issue, we conducted expert interviews to help identify potentially overlooked programmes. To assess why papers from SAVSNET did not appear in our literature search, we chose three papers from 2022 that originated from the SAVSNET [114–116]. Neither Green et al. nor Hall et al. have the concept cohort, longitudinal, or prospective in the title, abstract, author keyword, or as a mesh-term. For Petchell et al., the term cohort is included in the abstract, but it is more than five terms apart from the term dog. Therefore, with the included proximity operator in the search string, this paper was not retrieved.

The second limitation is that, the criteria used to decide on paper and also on cohort programme inclusion and exclusion were necessarily somewhat subjective; although we made strong efforts to objectify these by creating a formal list of six selection criteria questions, there is possibility that some papers were not captured in the mapping review. Another limitation is that for RQ 3–5, we relied on the available information extracted from the papers identified by the literature search. It is possible that some aspects of the programmes were not recorded in these papers and, therefore, were not included in the mapping review. Lastly, we fully acknowledge that we likely did not capture all ongoing and terminated cats and dogs cohort research programmes. The aim of the current work was not to identify a census of all programmes but to identify a sample of programmes from which useful inference could be derived.

Despite these listed limitations, we are confident that the current mapping review offers a useful overview of cat and dog cohort research programmes that can serve as a resource for those interested in undertaking similar research. Additionally, the mapping review provides some suggested learning from the existing programmes for their own future improvement.

## Conclusions

The mapping review identified 22 cats and dogs cohort programmes, predominantly in the US and the UK, with the majority focusing on dogs. Most programmes aimed to understand environmental and genetic factors influencing companion animal health. Digitalisation created new opportunities for more efficient data management, and especially for using veterinary electronic health records which were the most common data source in cats and dogs cohort programmes. The relative scarcity of veterinary cohorts compared to human medicine underscores the need for more robust efforts in this area, but poor reporting on planning and study design limits learning opportunities for future researchers. While scientific publications are effective for disseminating results related to their specific research question, they fall short in providing guidance for others on how to plan and conduct such cohort research programmes.

## Supporting information

**S1 File. Search string and keywords used in Rayya in the mapping review of cats and dogs research cohort programmes.**
(DOCX)

**S2 File. Data collection sheet used in the mapping review of cats and dogs research cohort programmes.** The file includes a codebook used for data collection, an Extraction sheet (empty), and a List of records that entered stage II of the selection.
(XLSX)

**S3 File. List of the veterinary research specialty(s) according to the Textbook of Small Animal Internal Medicine"by Stephen J. Ettinger and Edward C. Feldman.**
(DOCX)

**S1 Fig. Year of enrolment start and first publication in cats and dogs research cohort programmes included in the mapping review.**
(PDF)

**S2 Fig. Graph presenting number of data sources and the number of promotion methods in cats and dogs research cohort programmes included in the mapping review.**
(PDF)

**S1 Table. Detailed age inclusion/exclusion criteria across 19 programmes in the mapping review of cats and dogs cohort research programmes.** This table shows the age limit used for inclusion or exclusion in the programs. For 9 it was a criterion and for another 5 it was not a criterion whether the animal could participate in the program or not. However, for 3 programs, there was no information available regarding whether age was a criterion.
(DOCX)

**S2 Table. Considered outcomes in cat and dog cohort research programmes.**
(DOCX)

## Acknowledgments

We thank Christine Verhoustraeten for her contribution to constructing the search strategy. We also thank the Centre for Reproducible Science at the University of Zurich (UZH) for their support and resources.

## Author contributions

**Conceptualization:** Jessica Irene Landolt, Dan G O'Neill, Stefan Unterer, Sonja Hartnack, Malwina Ewa Kowalska.

**Data curation:** Jessica Irene Landolt, Malwina Ewa Kowalska.

**Formal analysis:** Malwina Ewa Kowalska.

**Methodology:** Dan G O'Neill, Sonja Hartnack, Malwina Ewa Kowalska.

**Project administration:** Jessica Irene Landolt, Malwina Ewa Kowalska.

**Resources:** Stefan Unterer.

**Software:** Jessica Irene Landolt, Malwina Ewa Kowalska.

**Supervision:** Malwina Ewa Kowalska.

**Visualization:** Malwina Ewa Kowalska.

**Writing – original draft:** Jessica Irene Landolt, Malwina Ewa Kowalska.

**Writing – review & editing:** Dan G O'Neill, Stefan Unterer, Sonja Hartnack, Malwina Ewa Kowalska.

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
