## [Decision Letter · Decision Letter 0]

16 Dec 2024

PONE-D-24-46436A Mapping review of worldwide current and previous cohort research programmes in cats and dogsPLOS ONE

Dear Dr. Kowalska,

Thank you for submitting your manuscript to PLOS ONE. After careful consideration, we feel that it has merit but does not fully meet PLOS ONE’s publication criteria as it currently stands. Therefore, we invite you to submit a revised version of the manuscript that addresses the points raised during the review process.

Please submit your revised manuscript by Jan 30 2025 11:59PM. If you will need more time than this to complete your revisions, please reply to this message or contact the journal office at plosone@plos.org . Please include the following items when submitting your revised manuscript:

We look forward to receiving your revised manuscript.

Kind regards,

Joshua Kamani, PhD

Academic Editor

PLOS ONE

Journal Requirements: When submitting your revision, we need you to address these additional requirements. 1. Please ensure that your manuscript meets PLOS ONE's style requirements, including those for file naming. The PLOS ONE style templates can be found at https://journals.plos.org/plosone/s/file?id=wjVg/PLOSOne_formatting_sample_main_body.pdf and https://journals.plos.org/plosone/s/file?id=ba62/PLOSOne_formatting_sample_title_authors_affiliations.pdf 2. When completing the data availability statement of the submission form, you indicated that you will make your data available on acceptance. We strongly recommend all authors decide on a data sharing plan before acceptance, as the process can be lengthy and hold up publication timelines. Please note that, though access restrictions are acceptable now, your entire data will need to be made freely accessible if your manuscript is accepted for publication. This policy applies to all data except where public deposition would breach compliance with the protocol approved by your research ethics board. If you are unable to adhere to our open data policy, please kindly revise your statement to explain your reasoning and we will seek the editor's input on an exemption. Please be assured that, once you have provided your new statement, the assessment of your exemption will not hold up the peer review process. 3. Please include captions for your Supporting Information files at the end of your manuscript, and update any in-text citations to match accordingly. Please see our Supporting Information guidelines for more information: http://journals.plos.org/plosone/s/supporting-information. 4. We note that there is identifying data in the Supporting Information file Supplementary File 2.xlsx. Due to the inclusion of these potentially identifying data, we have removed this file from your file inventory. Prior to sharing human research participant data, authors should consult with an ethics committee to ensure data are shared in accordance with participant consent and all applicable local laws. Data sharing should never compromise participant privacy. It is therefore not appropriate to publicly share personally identifiable data on human research participants. The following are examples of data that should not be shared: -Name, initials, physical address-Ages more specific than whole numbers-Internet protocol (IP) address-Specific dates (birth dates, death dates, examination dates, etc.)-Contact information such as phone number or email address-Location data-ID numbers that seem specific (long numbers, include initials, titled “Hospital ID”) rather than random (small numbers in numerical order) Data that are not directly identifying may also be inappropriate to share, as in combination they can become identifying. For example, data collected from a small group of participants, vulnerable populations, or private groups should not be shared if they involve indirect identifiers (such as sex, ethnicity, location, etc.) that may risk the identification of study participants. Additional guidance on preparing raw data for publication can be found in our Data Policy (https://journals.plos.org/plosone/s/data-availability#loc-human-research-participant-data-and-other-sensitive-data) and in the following article: http://www.bmj.com/content/340/bmj.c181.long. Please remove or anonymize all personal information (<Author names>), ensure that the data shared are in accordance with participant consent, and re-upload a fully anonymized data set. Please note that spreadsheet columns with personal information must be removed and not hidden as all hidden columns will appear in the published file. 5. Please review your reference list to ensure that it is complete and correct. If you have cited papers that have been retracted, please include the rationale for doing so in the manuscript text, or remove these references and replace them with relevant current references. Any changes to the reference list should be mentioned in the rebuttal letter that accompanies your revised manuscript. If you need to cite a retracted article, indicate the article’s retracted status in the References list and also include a citation and full reference for the retraction notice.

**Additional Editor Comments:**

Dear Authors

Kindly address the comments of the reviewers to unable us make final decision on the MS

Reviewers' comments:

Reviewer's Responses to Questions

**Comments to the Author**

1. Is the manuscript technically sound, and do the data support the conclusions?

Reviewer #1: Yes

2. Has the statistical analysis been performed appropriately and rigorously? 

Reviewer #1: N/A

3. Have the authors made all data underlying the findings in their manuscript fully available?

Reviewer #1: Yes

4. Is the manuscript presented in an intelligible fashion and written in standard English?

Reviewer #1: Yes

5. Review Comments to the Author

Reviewer #1: The article provides a systematic overview of cohort research programs involving cats and dogs. Using a mapping review methodology, the study identifies 22 programmes, primarily in the UK and US, focusing on topics like preventive medicine, geriatrics, and animal behavior. Key findings highlight gaps in study design reporting, funding transparency, and data management practices. The review emphasizes the need for better reporting standards, pre-registration practices, and increased collaboration to improve the quality and global reach of veterinary cohort research. The article also reports that the majority of cohort research programs (approximately 50%) are devoted to preventive medicine, although this detail is not explicitly mentioned in the article’s summary. Including this percentage, would enhance the clarity of findings. For instance, reporting that 11 out of 22 cohort programs focused on preventive medicine highlights this key insight more transparently.

Additionally, the main conclusions of these cohort research programs could be summarized to reflect common goals such as disease prevention, monitoring of health outcomes, and understanding environmental and genetic factors influencing companion animal health. Including such conclusions in the summary would provide a more comprehensive overview of the research's contributions to veterinary medicine.

On the other hand, these small set of observtions:

L123-L125: Could you please improve the syntax of these lines?

L127. In the paper context what does it mean demographics?

L209. Term "guardian of pets" is increasingly used instead of "owner" as it emphasizes responsibility, care, and protection. It aligns with a more compassionate and ethical approach to human-animal relationships.

6. PLOS authors have the option to publish the peer review history of their article (what does this mean? ). If published, this will include your full peer review and any attached files.

**Do you want your identity to be public for this peer review?** For information about this choice, including consent withdrawal, please see our Privacy Policy .

Reviewer #1: No

---

## [Author Response · Author response to Decision Letter 1]

16 Feb 2025

1) Ensure that your manuscript meets PLOS ONE's style requirements

- Thank you. Manuscript was adjusted to the PLOS One Style.

2) Please note that, though access restrictions are acceptable now, your entire data will need to be made freely accessible if your manuscript is accepted for publication. This policy applies to all data except where public deposition would breach compliance with the protocol approved by your research ethics board. If you are unable to adhere to our open data policy, please kindly revise your statement to explain your reasoning and we will seek the editor's input on an exemption. Please be assured that, once you have provided your new statement, the assessment of your exemption will not hold up the peer review process.

- Thank you for your comment. That was a mistake on my side. Data was and is free available at the OSF, which is indicated in the Acknowledgments section: “The data collection template is available in S2 File and at the OSF; data used for all analyses, analytic R code, and screener instructions are available at the OSF (https://osf.io/wkg53/).”

3) Please include captions for your Supporting Information files at the end of your manuscript, and update any in-text citations to match accordingly. Please see our Supporting Information guidelines for more information: http://journals.plos.org/plosone/s/supporting-information.

- Thank you. A supporting information section was included. Cytation and naming were done according to the PLOS One guidelines.

4) We note that there is identifying data in the Supporting Information file Supplementary File 2.xlsx. Due to the inclusion of these potentially identifying data, we have removed this file from your file inventory. Prior to sharing human research participant data, authors should consult with an ethics committee to ensure data are shared in accordance with participant consent and all applicable local laws

- Thank you very much for a detailed information. Authors names and Institution location was removed from a list of records that entered stage II of the selection in the mapping review.

5) Please review your reference list to ensure that it is complete and correct

- The list was reviewed and formatted. No cited papers have been retreated or added to the list.

Reviewer 1

1) The article provides a systematic overview of cohort research programs involving cats and dogs. Using a mapping review methodology, the study identifies 22 programmes, primarily in the UK and US, focusing on topics like preventive medicine, geriatrics, and animal behavior. Key findings highlight gaps in study design reporting, funding transparency, and data management practices. The review emphasizes the need for better reporting standards, pre-registration practices, and increased collaboration to improve the quality and global reach of veterinary cohort research. The article also reports that the majority of cohort research programs (approximately 50%) are devoted to preventive medicine, although this detail is not explicitly mentioned in the article’s summary. Including this percentage, would enhance the clarity of findings. For instance, reporting that 11 out of 22 cohort programs focused on preventive medicine highlights this key insight more transparently.

- Thank you for the comment. We expanded the abstract text to highlight and include additional information about the focus of identified cohort programmes. Line 42-43

2) Additionally, the main conclusions of these cohort research programs could be summarized to reflect common goals such as disease prevention, monitoring of health outcomes, and understanding environmental and genetic factors influencing companion animal health. Including such conclusions in the summary would provide a more comprehensive overview of the research's contributions to veterinary medicine

- Thank you for this valuable suggestion. We have adjusted the conclusions section to summarize the common goals of the cohort research programmes, including understanding environmental and genetic factors influencing companion animal health. Line 724-725

3) L123-L125: Could you please improve the syntax of these lines?

- Thank you for your suggestion to improve the syntax. We have revised the sentence for clarity as follows:

'In this mapping review, we specifically focused on cohort research programs involving only cats or dogs, which we will refer to as "programmes." Line 132-134

L127.

4) In the paper context what does it mean demographics?

- Thank you for your comment. To make it more cleare we add more examples: “In the eligible programmes, animals are enrolled primarily based on demographics (e.g., cat or dog species, age, sex, breed) rather than health status (e.g., cancer registries)” Line 137

5) L209. Term "guardian of pets" is increasingly used instead of "owner" as it emphasizes responsibility, care, and protection. It aligns with a more compassionate and ethical approach to human-animal relationships.

- Thank you for the valuable comment. We left “owner” in the phrase “owner consent forms” as we believed this is still a commonly used term and functioning as a whole phrase. In any other places, text was adjusted according to your suggestion. Line 189, 226, 237, 426, 440, 482, 486, 488, 515, 526, 528

---

## [Editor Report · Decision Letter 1]

28 Feb 2025

A Mapping review of worldwide current and previous cohort research programmes in cats and dogs

PONE-D-24-46436R1

Dear Dr. Kowalska,

We’re pleased to inform you that your manuscript has been judged scientifically suitable for publication and will be formally accepted for publication once it meets all outstanding technical requirements.

Kind regards,

Joshua Kamani, PhD

Academic Editor

PLOS ONE
---

## [Editor Report · Acceptance letter]

PONE-D-24-46436R1

PLOS ONE

Dear Dr. Kowalska,

I'm pleased to inform you that your manuscript has been deemed suitable for publication in PLOS ONE. Congratulations! Your manuscript is now being handed over to our production team.

Kind regards,

on behalf of

Dr. Joshua Kamani

Academic Editor

PLOS ONE